# Comparison between Olympic Weightlifting Lifts and Derivatives for External Load and Fatigue Monitoring

**DOI:** 10.3390/healthcare10122499

**Published:** 2022-12-10

**Authors:** Joaquim Paulo Antunes, Rafael Oliveira, Victor Machado Reis, Félix Romero, João Moutão, João Paulo Brito

**Affiliations:** 1Sports Science School of Rio Maior—Polytechnic Institute of Santarém, 2040-413 Rio Maior, Portugal; 2Life Quality Research Centre, 2040-413 Rio Maior, Portugal; 3Federação de Halterofilismo de Portugal, 2835-104 Moita, Portugal; 4Research Centre in Sports Sciences, Health Sciences and Human Development, 5001-801 Vila Real, Portugal; 5Sport Sciences Department, University of Trás-os-Montes e Alto Douro, 5001-801 Vila Real, Portugal

**Keywords:** Clean and Jerk, load monitoring, Olympic exercises, Power Clean, power, Snatch, Squat, weightlifting derivatives

## Abstract

Load management is an extremely important subject in fatigue control and adaptation processes in almost all sports. In Olympic Weightlifting (OW), two of the load variables are intensity and volume. However, it is not known if all exercises produce fatigue of the same magnitude. Thus, this study aimed to compare the fatigue prompted by the Clean and Jerk and the Snatch and their derivative exercises among male and female participants, respectively. We resorted to an experimental quantitative design in which fatigue was induced in adult individuals with weightlifting experience of at least two years through the execution of a set of 10 of the most used lifts and derivatives in OW (Snatch, Snatch Pull, Muscle Snatch, Power Snatch, and Back Squat; Clean and Jerk, Power Clean, Clean, High Hang Clean, and Hang Power Clean). Intensity and volume between exercises were equalized (four sets of three repetitions), after which one Snatch Pull test was performed where changes in velocity, range of motion, and mean power were assessed as fatigue measures. Nine women and twelve men participated in the study (age, 29.67 ± 5.74 years and 28.17 ± 5.06 years, respectively). The main results showed higher peak velocity values for the Snatch Pull test when compared with Power Snatch (*p* = 0.008; ES = 0.638), Snatch (*p* < 0.001; ES = 0.998), Snatch Pull (*p* < 0.001, ES = 0.906), and Back Squat (*p* < 0.001; ES = 0.906) while the differences between the Snatch Pull test and the derivatives of Clean and Jerk were almost nonexistent. It is concluded that there were differences in the induction of fatigue between most of the exercises analyzed and, therefore, coaches and athletes could improve the planning of training sessions by accounting for the fatigue induced by each lift.

## 1. Introduction

Olympic Weightlifting (OW) is a dynamic strength and power sport in which two complex lifts/exercises are performed in competition: the Snatch and the Clean and Jerk (C&J). During these lifts, weightlifters have achieved some of the highest peak power outputs reported in the literature [1,2].

The Snatch requires a weighted barbell to be lifted from the floor (usually using a wide grip) to an overhead position in one continuous movement [3]. The C&J is divided into two main phases, in which the first requires the barbell to be raised from the floor (using a shoulder-width grip) to the front of the shoulders in one continuous movement [4], and the second phase consists of a jerk, in which the barbell is propelled from the shoulders to arm’s length overhead by forces produced primarily by the hips and thighs [5].

Considering that weightlifting is used in strength development in most sports, some questions arise: how to train with weightlifting efficiently and how it reflect in the performance of other sports that use weightlifting exercises and their derivatives (i.e., variations that omit part of the full lift such as the Hang Clean, Hang Snatch, Power Clean, Power Snatch, and High pull). The answer to these questions may require enhancing the testing and training methods of weightlifting with a combination of the main exercises and their derivatives [6]. Weightlifting exercises and their derivative exercises have become a popular training modality to improve high strength and power expressions throughout the whole force–velocity spectrum during movement, across a range of sports [7,8,9,10,11,12,13].

Monitoring, planning, and periodizing training loads are critical factors when it comes to the athlete’s development and progression. There has been an attempt by researchers to increasingly identify the variables of training and to control them. In fact, in the past, even the successful Bulgarian methodology tried to reduce some variables by reducing the variety of exercises used [14]. Therefore, there are still some factors that remain unknown. In OW training, load variables such as volume (number of repetitions multiplied by the number of sets) are often manipulated. On the other hand, intensity is expressed relative to the maximal load (kilograms) obtained in the main exercises. Another variable that is also commonly used is the total load, which is characterized by the number of sets multiplied by the number of repetitions multiplied by the kilograms lifted, also known as tonnage [15].

The magnitude of force production and the capacity to perform a given amount of work as rapidly as possible are often suggested as the primary underpinning qualities of sports skills. Thus, developing strength, power, and speed capabilities is frequently the primary aim of many athletic development programs [16]. Despite the variables that define the load being described in terms of intensity by volume [17], there are several parallel factors that may still be associated with this quantification, namely, the type and exercise selection [18]. More recently, other algorithms for the grouping and selection of exercises have been proposed, in some cases based on technical efficiency [19]. Factors such as the number and type of muscle fibers involved, either because of the complexity of the movement or because of the amount of force developed in a given unit of time, can vary in each exercise [11], thus creating an unknown amount of additional fatigue. Nonetheless, in strength training, the external load is related to the external resistance (load) lifted, but it can also be related to the work completed or the velocity achieved during exercises [20].

Several researchers [21,22,23] have highlighted strong relationships between load and movement velocity fatigue, with the assessment of strength qualities being load–velocity specific. In fact, previous studies have confirmed that the speed of movement provides a determinant of the level of effort during resistance training as well as a variable of the degree of fatigue [24,25]. Therefore, it is particularly important to know the fatigue induced by the different OW derivatives when programming the training load. It is essential to know which exercises induce greater fatigue and its magnitude. High-power outputs and the rate of force development expressed in weightlifting movements and derivatives [2], in conjunction with the motor control and coordination demands on the trunk and lower body muscles to stabilize and transmit forces [26], can effectively impact and compromise various aspects of an athlete’s load–velocity profile [16].

This is a topic that has been scarcely addressed in the literature, which lacks results that could improve coaching, both in terms of exercise selection and the distribution of exercise along microcycles, mesocycles, and macrocycles. Several attempts have already been made to try to organize the various exercises into the clusters approach [15]. However, exercise-induced fatigue has never been investigated.

Thus, the aim of the present study was to compare the external load and fatigue prompted by the Clean and Jerk, the Snatch, and their derivative exercises (Snatch, Snatch Pull, Muscle Snatch, Power Snatch, and Back Squat; Clean and Jerk, Power Clean, Clean, High Hang Clean, and Hang Power Clean) among male and female participants, respectively. It was hypothesized that when volume and intensity are equated, there are differences in external load and fatigue induced by performing the different OW derivative exercises.

## 2. Materials and Methods

### 2.1. Design

This was a cross-sectional study conducted over two separate days, set apart by a minimum of three days and a maximum of five days. All procedures were recorded for future consultation at the protocols.io website (accessed on 1 January 2020.), and the sample represents more than 10% of the OW Portuguese population [27].

### 2.2. Participants

A priori power analysis using G*Power (Statistical Power Analyses software for Windows—RRID: SCR_013726) was completed [28]. A sample size calculation was made for the difference between two dependent means (paired sample t-test), an effect size of 0.8, an alpha of ≤0.05, and a beta of 0.95. It was determined that at least 19 participants were needed. Twenty-one Caucasian participants, twelve males and nine females, volunteered to participate in the study.

The inclusion criteria were to be aged between 18 and 40 years; having more than 2 years of OW training; competing at the national level; and having between 61 and 96 kgs of bodyweight for the male group and between 49 and 71 kgs for the female group. The characteristics of the participants are presented in Table 1.

Data collection took place at each participant’s usual training gym. Prior to their participation, each participant was familiarized with all procedures. Moreover, they read and signed a written informed consent form, in accordance with the university’s institutional review board, before data collection. This study was designed according to the recommendations of the World Medical Association’s Declaration of Helsinki of 1975, as revised in 2013, for human studies and approved by the Institutional Ethics Committee (approval number: 07A-2021ESDRM).

### 2.3. Exercise Selection

The rationale for the exercise selection was based on three factors: its ability to enhance the force–velocity profile of athletes [12]; the ability of each derivative to serve as a foundational exercise that enables the progression to more complex weightlifting movements [11,12]; and the exercise frequency applied by OW coaches [29]. The selected exercises were the Snatch and its derivative exercises (Muscle Snatch; Power Snatch; Snatch; Snatch Pull; Back Squat) and the Clean and Jerk (C&J) and its derivative exercises (Power Clean; C&J; Clean; High Hang Clean; Hang Power Clean).

### 2.4. External Load and Fatigue Assessment

Usually, the isometric mid-thigh pull test (IMTP) is a reliable and popular way to test maximal strength in adult athletes. Administering a partial movement test is a safer and more time-efficient method than traditional 1RM testing. The IMTP produces relatively little fatigue and has a low potential for injury [30], but it proved to be less effective in predicting the competitive performance of OW than other tests [31]. When considering the concept of neuromuscular fatigue, it is important to note that isometric versus dynamic measurements do not provide the same results. Additionally, the bar’s range of motion (ROM) also plays an important role in OW, and it seems to be an important factor when assessing fatigue [32]. Therefore, we opted for the Snatch Pull test (SPT) as a reference measure, which has been correlated with the personal record (PR) of the Snatch exercise (r = 0.99) [33]. In all OW derivatives, the external load variables of mean velocity, peak velocity, mean power, and ROM were measured using the Isoinertial Dynamometer Vitruve (Vitruve encoder; Madrid, Spain) (previously, Speed4Lifts) [34]. Both mean and peak velocities were considered based on the measurement of fatigue according to previous references [24,25]. Moreover, this type of test can regularly be applied during weightlifting training as a valid alternative to the personal record Snatch test to assess individualized progression in weightlifting performance over time [33].

Since all these lifts have correlation intensity with each other, and Muscle Snatch is referenced as 60 to 65% of the Snatch PR, an intensity load of 60% was chosen. Therefore, setting it as the baseline intensity, the volume chosen (4 sets of 3 repetitions) was the amount of load that is usually performed by lifters within the intensity already settled upon [29].

### 2.5. Procedures

On the first day of data collection, participants started early in the morning for an anthropometric assessment, namely, height, weight, and body composition using bioimpedance analysis.

#### 2.5.1. Anthropometric and Body Composition Assessment

The anthropometric and body composition measurements were obtained with the subjects dressed in light clothing without shoes following previous recommendations [35] using a stadiometer with an incorporated scale (Seca 220, Hamburg, Germany) according to standardized procedures [36]. The body composition data were obtained with bioelectrical impedance analysis using Inbody S10 (model JMW140, Biospace Co, Ltd., Seoul, Korea), according to the manufacturer’s guidelines [37,38]. Eight electrodes were placed on eight tactile points (thumbs, middle fingers of both hands, and the ankles of both feet) to perform a multi-segmental frequency analysis. The parameters collected were body fat mass (BFM) and fat-free mass (FFM).

The measurements were carried out in the morning in a room with an ambient temperature and relative humidity of 22–23 °C and 50–60%, respectively, after a minimum of 8 h of fasting and after the bladder was emptied, following previous suggestions [35,39]. The participants adopted a supine position with their arms and legs abducted at a 45° angle; the skin was cleaned with ethyl alcohol and hydrophilic cotton at the eight electrode placement sites. After a 10 min rest in a room without noise, eight electrodes were placed on the cleaned surfaces, and the measurements were performed.

Before data collection, participants did not exercise or ingest caffeine or alcohol during the 12 h prior to the assessment. In addition, participants removed all objects that could interfere with the bioelectrical impedance assessment.

Female participants were only assessed if they were in the luteal phase of ovulatory menstrual cycles. Otherwise, they waited until they were in the luteal phase. All the assessments were performed by the same evaluator to minimize possible measurement errors [40].

#### 2.5.2. Test Protocol

After anthropometric and body composition assessments, an explanation of the protocol was provided. A 10 min warm-up, including mobility exercises, OW repetitions, and jumps, was carried out before the beginning of each training session. To minimize the risk of injury, there were always two assistants to monitor exercise execution.

Participants started their personal warm-up exercises/specific-for-training session: up to 60% of the Snatch 1RM followed by two 50%, one 70%, and one 100% Snatch 1RM and SPT attempts separated by 1 min of recovery [41]. Before each SPT, verbal feedback cues were given by coaches in a standardized form, namely, “Pull hard and fast”.

On the first test day, the Snatch and derivative exercises protocol took place. After the warm-up, the baseline SPT evaluation occurred (Figure 1), making a 1RM Snatch of personal record, after which, data were collected. Then, participants rested for 1 min, followed by a Muscle Snatch protocol of 4 sets of 3 repetitions at 60% of the Snatch 1RM (1 min rest between sets). After the protocol, participants then took a 1 min rest before the post Muscle Snatch SPT evaluation (1RM).

This was followed by a Power Snatch protocol of 4 sets of 3 repetitions at 60% of the Snatch 1RM (1 min rest between sets). After the protocol, participants then took a 1 min rest before the post Power Snatch SPT evaluation, followed by one repetition at 100% Snatch 1RM, after which, data were collected; participants would then rest 1 min.

The same protocol was used for the Snatch, Snatch Pull, and Back Squat. On the second test day, three days after the tests were performed on the Snatch derivatives, the C&J and derivative exercises protocol was performed.

The same protocol used in the Snatch derivatives was used for all C&J derivatives in the following order: Power Clean, C&J, Clean, High Hang Clean, and Hang Power Clean. In this protocol, 60% of the C&J 1RM was used.

### 2.6. Statistical Analysis

All statistical analyses were performed using IBM SPSS for Windows (IBM Corp. Released 2020., Version 28.0. Armonk, NY, USA). The data are described as mean ± standard deviation (SD), standard error of the mean (SEM), and mean difference (MD) with a 95% confidence interval (CI). The Shapiro–Wilk test was used for testing normality. Differences between exercises were examined using a paired samples t-test (velocity, range of motion, and mean power within each exercise monitored using the isoinertial dynamometer). An a priori level of significance was set at *p* < 0.05 and a percentage change with a 95% CI. The effect size (ES) was calculated to determine the magnitude of the effects through Cohen’s d (by the difference of two pairs of means, which are then divided by the standard deviation from the data), and the following thresholds were applied: large d, > 0.8; moderate d, between 0.8 and 0.5; small d, between 0.49 and 0.20; trivial d, < 0.2 [42].

## 3. Results

### 3.1. Snatch Derivative Protocols

Analyzing the mean power for the entire sample (*n* = 21), it was found that, after the Muscle Snatch protocol, there were no significant differences while, post-Power Snatch, -Snatch, -Snatch Pull, and -Back Squat showed a significant difference (Table 2). However, when considering the gender group analyses separately, the female group (*n* = 9) reveals no difference after the Muscle Snatch and Power Snatch protocols, whereas Snatch, Snatch Pull, and Back Squat manifested a significant difference. The male group (*n* = 12) did not reveal significant differences in mean power for any exercise.

Mean velocity evidenced a significant difference in the Snatch and Back Squat (Table 2) for the total sample. When the gender groups were analyzed, the female group showed differences after the Snatch, Snatch Pull, and Back Squat (Table 3). No differences were found in the male group.

Peak velocity did not show a significant difference in the Muscle Snatch, while the remaining derivatives showed significant differences (Table 2). The female group did not report differences after the Muscle Snatch and Power Snatch protocols (Table 3). No differences were found for the male group in the Muscle Snatch and Back Squat.

For the total sample, only the Muscle Snatch protocol revealed differences in the range of motion (Table 2). In the gender analysis (Table 3), the female group revealed that, after the Muscle Snatch protocol, the Snatch, Snatch Pull, and Back Squat exercises presented a significant difference, while in the male group, only the Snatch Pull showed differences.

In the assessment of the Snatch variables, it was verified that the ROM and post-Snatch Pull protocol, as well as the peak velocity, post-Snatch, -Snatch Pull, and -Back Squat, showed differences when the total sample was analyzed.

### 3.2. Clean and Jerk Derivative Protocols

Differences in the C&J mean power and mean velocity were only found when considering the whole sample (Table 4) and when considering the male group (Table 5).

Regarding peak velocity, no differences were found in any of the exercises or in either group. For the total sample, ROM only showed a significant difference in the C&J (Table 4). In the group analysis, only the male group showed differences in the C&J and Clean (Table 5).

## 4. Discussion

The aim of the present study was to compare the external load and fatigue prompted by the Clean and Jerk, the Snatch, and their derivative exercises (Snatch, Snatch Pull, Muscle Snatch, Power Snatch, and Back Squat; Clean and Jerk, Power Clean, Clean, High Hang Clean, and Hang Power Clean) among male and female participants, respectively. The majority of these exercises are used in OW, as well as in general strength and conditioning training programs for various sports [9,43,44,45,46]. It was hypothesized that when volume and intensity are equated there are differences between external load and fatigue induced by different OW exercises.

The main results showed that, for the total sample, significant differences were found in the Snatch Pull, Snatch, and Back Squat ROM and on the C&J ROM. Regarding the mean power, significant differences were found in the Power Snatch, Snatch, Snatch Pull, Back Squat, and C&J. Regarding peak velocity, significant differences were found in the Power Snatch, Snatch, Snatch Pull, and Back Squat. Regarding the mean velocity, significant differences were found in the Snatch Pull and Back Squat.

When genders were analyzed separately, the female group showed significant differences in the Snatch ROM, Snatch Pull, and Back Squat, while in the male group, differences were found in the ROMs of the Snatch Pull, C&J, and Clean. Regarding mean power, the female group presented significant differences in the Snatch, Snatch Pull, and Back Squat, while the male group showed significant differences in mean power in the C&J. The female group also revealed significant peak velocity differences in the Snatch, Snatch Pull, and Back Squat, while the male group revealed significant differences in the Power Snatch, Snatch, and Snatch pull. In addition, the female group showed significant differences in mean velocity in the Snatch, Snatch Pull, and Back Squat, while the male group only showed significant differences in the C&J. The fact that women can perform a greater number of intermittent contractions than men, even when the two groups are matched for strength, has been reported before [47], and the same effect may occur in OW training.

Considering the whole sample, almost all variables presented significant differences, as well as moderate-to-large effect size values. Peak velocity seems to present the most significant differences in both groups; however, in the female group, Snatch derivatives seem to show significant differences in every variable studied. This effect might be related to better technique proficiency and consistency in female lifters. On the other hand, the male group only showed significant differences in peak velocity. The fatigue induced by each exercise may be related to the individualized load–velocity relationship and to the specific characteristics of the participant [48,49]. Some studies [49,50] have reported that intersubject variability seems to be reduced when the loads are prescribed based on the individual load–velocity relationship. However, some coaches prefer to prescribe the loads to match a specific number of repetitions rather than using a prescription method based on bar velocity. Still, there is high intersubject variability between the number of repetitions performed and neuromuscular fatigue [51].

Some authors [11,12,17,21,23,33,49] have described a theoretical relationship between force and velocity with special consideration for weightlifting derivatives. The high-force end of the force–velocity curve features weightlifting derivatives that develop the largest forces due to the loads that can be used. As Suchomel et al. [52] point out, the proper implementation and progression of resistance training exercises throughout training facilitate the development of an athlete’s force–velocity profile [52,53], which has been cited as an important aspect of athletic performance [54,55]. Specifically, the biomechanical and physiological characteristics of each weightlifting derivative may indicate that certain derivatives should be prescribed during certain training phases to meet the training goals of each phase. Thus, information that may assist practitioners when it comes to programming exercises to optimally develop these characteristics would be beneficial. In the present study, the only exercise that did not show any difference in any variable was the Muscle Snatch, and this exercise was the one with the highest ROM.

A higher barbell ROM has a direct relationship with the subject’s height [11,12,56,57], meaning that if the lifter is taller, the barbell needs to have a higher displacement than if the lifter is shorter. OW is a competitive sport that requires athletes to lift a maximal amount of weight in the Snatch and C&J. OW’s main distinction in sports training is velocity, meaning that other sports mostly involve training to develop more speed, maintaining the load of the athlete (bodyweight in most cases). However, OW aims to maintain the ideal velocity for each exercise according to its height. Therefore, it is also correlated with the lifter’s height [56], and manipulating barbell weight could also indicate that OW lifters are more resistant to velocity loss than other kinds of athletes.

Recent research has also reported that different individual physical characteristics lead to different fatigue levels and recovery [57], and this could have led to greater variability in the study results. More than half of the participants showed increases in most variables instead of an expected decrease induced by fatigue. The post-activation potential effect might be involved in these findings, as this effect is a possible result of muscle contractions, and, utilized during subsequent explosive activity, it could potentially enhance power and, therefore, performance. However, while a previous effort might also induce fatigue, it is the balance between the post-activation potentiation effect and fatigue onset that will determine the effect of a previous effort on performance in an explosive movement. This relationship is affected by several variables, including volume and intensity and subject characteristics, as well as others [47,58]. Thus, it can also be inferred that some athletes probably did not quite induce this effect during their warm-ups. The fact that the warm-up was not standardized can be considered a limitation herein. In future studies, the warm-up should be controlled and also equalized among subjects since it may affect performance in explosive movements [59].

Additionally, we can speculate that some types of exercises may contribute more to the better potentiation of muscle contractions due to the lifted load, the force–velocity curve, and the different levels of induced fatigue [47,57,58,60,61,62]. The neuromuscular adaptations induced by weightlifting training strongly depend on the manipulation of strength training variables, such as the exercise type and sequence, load magnitude, volume, interset and intraset rest periods, and lifting velocity [63,64]. A common concern for coaches is deciding how much weight their athletes should lift in a particular exercise, as resistance-training-induced adaptations are highly dependent on the intensity used [65].

In addition to the manipulation of variables intrasession, coaches program exercises within periodized programs to vary the intensity of the training stimuli. Regarding squat movements, the exercise stimulus may be varied based on the depth and variation of the squat [66], as well as the load that is prescribed. As a result, the force–velocity characteristics of the training stimulus will be modified, but the athlete’s force–velocity profile may be fully developed. There was a report that certain weightlifting derivatives emphasize force or velocity more than others [12]. Thus, it seems that a sequential progression and combination of weightlifting derivatives can be beneficial to athletes when it comes to increasing force and power development rates. Moreover, techniques refined during earlier training phases may facilitate increases in the load used for each exercise.

Sports such as OW, along with its derivatives, require a single high-force or high-velocity effort. These movements typically involve a burst of concentric muscular activity in the agonist muscles, followed by a phase of relaxation, which, during the motion, continues due to stored momentum. This type of movement is also known as ballistic movement/action [67]. In voluntary muscle contractions, the total force output of a muscle depends primarily on the number of motor units and the firing frequency of those motor units, in which a higher force output will result in more motor units firing frequency [67]. In fact, motor unit recruitment is known to be a critical factor in maximal or ballistic contractions, as well as inducing fatigue. This principle—known to be the recruitment threshold of a motor neuron—can be directly related to the size of its axon. In other words, the larger the axon, the greater the amount of stimulation required [68]. In fact, there is some evidence of the selective activation of large motor units if the motor task readily demands those motor units [69]. Moreover, ballistic exercises elicit several acute and chronic neurological changes. The standard recruitment of motor units, according to the size principle, stays consistent at submaximal exercise intensities but appears to be violated in ballistic movements. It seems that the motor task, more than any other variable, determines the sequence of activation [67].

All OW exercises and their derivatives have relatively high motor recruitment. However, more complex exercises empirically require more units. Therefore, they are supposed to use more energy, leading to greater fatigue. The fact that some exercises did not show fatigue in the current study may be associated with the fact that the volume or intensity fatigue threshold was not met. Recording the bar velocity at which submaximal loads are lifted is a potential method of quantifying the load as a function of the fatigue it causes [24,70]. Researchers have reported the general relationship between lifting velocity and the %1RM in different exercises. Nowadays, it seems to be the consensus that the individualized load–velocity relationship allows for a better assessment of athlete fatigue, mainly because the %1RM–velocity relationship is subject-specific [51,71]. Unfortunately, little information exists regarding the possibility of predicting the number of repetitions from the recording of lifting velocity when powerlifting training (i.e., OW-derivative exercise) is used by strength and power athletes up until the final days prior to a competition. Therefore, understanding how different derivatives influence peak power performance is critical [31]. Recent evidence suggests that many coaches and support staff are taking an increasingly scientific approach to load monitoring [12,13,23,46,50,72].

Some limitations of the present study may be considered. The randomization of the sample could only be accomplished to a certain extent since the population to be studied is small in number by itself, and the inclusion criteria further narrow this choice. Therefore, the small sample could also be pointed to as a relative limitation because both males and females were analyzed as a group, and the samples were even smaller when the genders are separated. The fact that we compared parameters in women and men as one homogeneous group could be considered questionable and should be considered a study limitation. However, this methodological decision stems from a practical issue, in that trainers test both men and women together [6]. Moreover, specific warm-ups were not standardized, mainly because lifters have their own warm-up routines, which we choose not to interfere in. However, this may constitute another unaccountable variable that could have influenced the first and second SPT.

Future research should take the previous information into account and try to measure 1RM, for example, by determining the catch height of each lifter and then setting 1RM using their respective SPT height achievements.

Nonetheless, the main practical outcome of this study was adding the various relative fatigue values to the overall load, whether in form of a percentage or a fraction. For example, considering peak velocity in the female group, Snatch had a 71% (0.10 m/s) difference in relative fatigue, and the Back Squat had the maximum relative fatigue difference (0.14 m/s), meaning that, when we multiply for the same load, let us say (100 kgs) × (0.71), Snatch = 71 in terms of the relative fatigue, while for (100 kgs) × (0.100), the Back Squat = 100 in terms of the relative fatigue, meaning that, for the same load (volume × Intensity), the Back Squat will fatigue the athlete 30% more than the Snatch.

## 5. Conclusions

This intervention confirmed the study hypothesis: when volume and intensity are equated, there are differences between fatigue induced by various OW exercises.

In Snatch derivatives, peak velocity was a good variable to quantify fatigue in both genders, while in all other variables, it was only sensible in females. In addition, females seem more sensitive to fatigue in Snatch derivatives. Snatch derivatives are well known for their velocity-developing capability; therefore, fatigue may be explained more effectively using a test that mimics the movement itself, such as the SPT.

In C&J derivatives, females did not present statistically significant results; therefore, they showed that more volume and or intensity are needed to induce measurable fatigue. Regarding the male group, ROM seems to be the variable that we can better rely on, and, in addition, C&J derivative exercises are less velocity-dependent; this could explain ROM’s ability to quantify fatigue.

The ten exercises studied showed different external load and fatigue levels between them. However, it was not possible to quantify the magnitude of the different variables. This is likely the consequence of individual physiological adaptations and responses to exercise.

Coaches may plan according to these findings, specifically, as to C&J variables, by using higher relative loads in the exercises where fatigue was not found. Furthermore, using peak velocity in the Snatch and its derivatives plus ROM in the C&J and its derivatives seems to be best for training control in OW.

## Figures and Tables

**Figure 1 healthcare-10-02499-f001:**
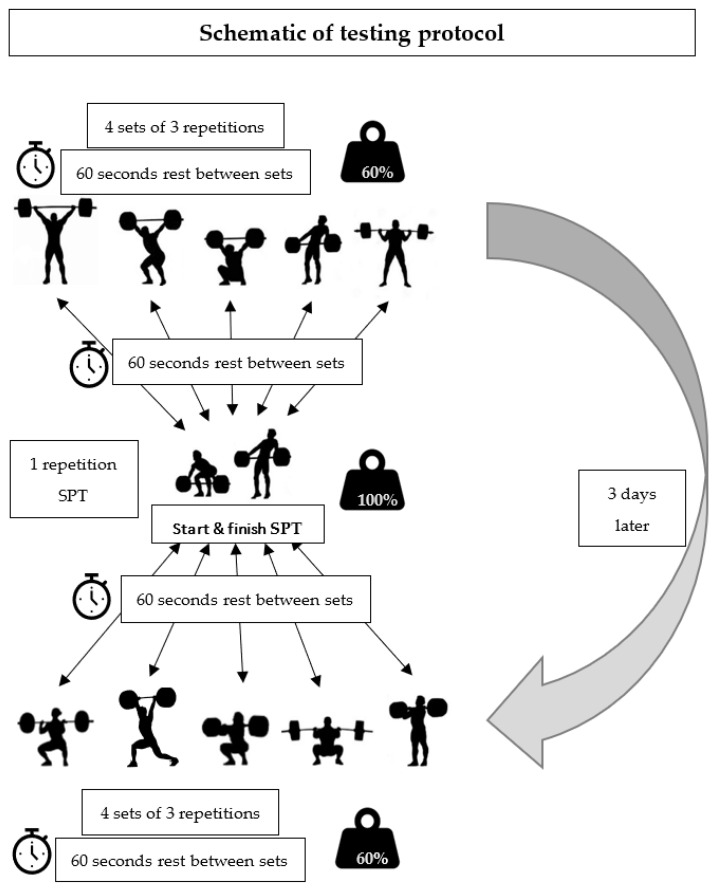
Testing protocol schematic.

**Table 1 healthcare-10-02499-t001:** Participants’ characteristics.

	Age (Years)	Height (cm)	Weight (kg)	BF (%)	FFM (kg)
**Female**	29.7 ± 5.7	158.8 ± 6.7	60.8 ± 7.3	17.8 ± 7.6	48.9 ± 7.7
**Male**	28.1 ± 5.0	174.5 ± 6.0	79.5 ± 5.3	17.0 ± 5.1	65.9 ± 5.0
**Total**	28.8 ± 5.3	167.8 ± 10.1	71.5 ± 11.2	17.3 ± 6.2	58.6 ± 10.6

BF, body fat; FFM, fat-free mass.

**Table 2 healthcare-10-02499-t002:** Baseline and post-values of the Snatch Pull test for the Snatch derivatives (♀♂ = 21).

Parameter	Weightlifting Derivative		Mean ± SD	SEM	MD (95% CI)	*p* (ES)
**ROM** **(cm)**	SPT		Baseline			
			106.49 ± 7.49	1.64		
			Post			
	Pair 1	Muscle Snatch	107.33 ± 7.75	1.69	−0.85 (−2.65; 0.95)	0.338 (−0.214)
	Pair 2	Power Snatch	105.15 ± 7.93	1.73	1.34 (−0.36; 3.04)	0.116 (0.358)
	Pair 3	Snatch	104.19 ± 7.85	1.71	2.30 (0.87; 3.73)	0.003 * (0.731)
	Pair 4	Snatch Pull	102.82 ± 8.63	1.88	3.67 (1.97; 5.36)	<0.001 ** (0.986)
	Pair 5	Back Squat	103.97 ± 9.41	2.05	2.52 (0.42; 4.62)	0.021 * (0.547)
**Mean Power** **(w)**	SPT		Baseline			
			706.55 ± 187.58	40.93		
			Post			
	Pair 1	Muscle Snatch	701.93 ± 189.80	41.42	4.61 (−18.41; 27.64)	0.680 (0.091)
	Pair 2	Power Snatch	681.19 ± 181.14	39.53	25.36 (0.93; 49.79)	0.043 * (0.472)
	Pair 3	Snatch	677.11 ± 183.49	40.04	29.44 (0.32; 58.55)	0.048 * (0.460)
	Pair 4	Snatch Pull	664.41 ± 180.76	39.44	42.14 (15.84; 68.44)	0.003 * (0.729)
	Pair 5	Back Squat	671.32 ± 190.58	41.59	35.22 (10.03; 60.42)	0.009 * (0.636)
**Peak Velocity** **(m/s)**	SPT		Baseline			
			1.81 ± 0.17	0.04		
			Post			
	Pair 1	Muscle Snatch	1.78 ± 0.18	0.04	0.04 (−0.01; 0.09)	0.125 (0.350)
	Pair 2	Power Snatch	1.76 ± 0.19	0.04	0.06 (0.02; 0.10)	0.008 * (0.638)
	Pair 3	Snatch	1.73 ± 0.17	0.04	0.08 (0.05; 0.12)	<0.001 ** (0.998)
	Pair 4	Snatch Pull	1.72 ± 0.15	0.03	0.09 (0.05; 0.14)	<0.001 ** (0.906)
	Pair 5	Back Squat	1.73 ± 0.18	0.04	0.08 (0.04; 0.13)	<0.001 ** (0.906)
**Mean Velocity** **(m/s)**	SPT		Baseline			
			0.94 ± 0.13	0.03		
			Post			
	Pair 1	Muscle Snatch	0.93 ± 0.12	0.03	0.01 (−0.02; 0.04)	0.508 (0.147)
	Pair 2	Power Snatch	0.91 ± 0.13	0.03	0.03 (0.00; 0.06)	0.050 (0.455)
	Pair 3	Snatch	0.90 ± 0.13	0.03	0.04 (0.00; 0.07)	0.030 * (0.509)
	Pair 4	Snatch Pull	0.92 ± 0.15	0.03	0.02 (−0.06; 0.10)	0.604 (0.115)
	Pair 5	Back Squat	0.89 ± 0.13	0.03	0.05 (0.02; 0.08)	0.003 * (0.727)

SPT, Snatch Pull test; ROM, range of motion; *, *p* < 0.05; **, *p* < 0.001; SD, standard deviation; SEM, standard error of the mean; MD, mean difference; CI, confidence intervals; ES, effect size.

**Table 3 healthcare-10-02499-t003:** Baseline and post-values of Snatch Pull test for the Snatch derivatives based on gender (♀ = 9; ♂ = 12).

Parameter		Weightlifting Derivative		Mean ± SD	SEM	MD (95% CI)	*p* (ES)
**ROM (cm)**	**Female**	SPT		Baseline			
				105.22 ± 8.25	2.75		
				Post			
		Pair 1	Muscle Snatch	105.16 ± 9.00	3.00	0.07 (−2.18; 2.32)	0.947 (0.023)
		Pair 2	Power Snatch	104.58 ± 9.29	3.10	0.64 (−1.95; 3.23)	0.585 (0.189)
		Pair 3	Snatch	102.04 ± 9.17	3.06	3.18 (1.17; 5.18)	0.006 * (1.218)
		Pair 4	Snatch Pull	100.40 ± 10.44	3.48	4.82 (2.73; 6.91)	0.001 * (1.776)
		Pair 5	Back Squat	100.03 ± 10.91	3.64	5.19 (2.48; 7.90)	0.002 * (1.474)
	**Male**	SPT		Baseline			
				107.43 ± 7.10	2.05		
				Post			
		Pair 1	Muscle Snatch	108.97 ± 6.59	1.90	−1.53 (−4.45; 1.38)	0.272 (−0.334)
		Pair 2	Power Snatch	105.57 ± 7.16	2.10	1.87 (−0.71; 4.44)	0.139 (0.460)
		Pair 3	Snatch	105.79 ± 6.66	1.92	1.64 (−0.55; 3.84)	0.128 (0.475)
		Pair 4	Snatch Pull	104.63 ± 6.91	2.00	2.80 (0.11; 5.49)	0.042 * (0.663)
		Pair 5	Back Squat	106.91 ± 7.23	2.09	0.52 (−2.28; 3.32)	0.692 (0.117)
**Mean Power (w)**	**Female**	SPT		Baseline			
				557.79 ± 128.94	42.98		
				Post			
		Pair 1	Muscle Snatch	540.79 ± 113.25	37.75	17.00 (−0.81; 34.81)	0.059 (0.734)
		Pair 2	Power Snatch	536.32 ± 121.34	40.45	21.47 (−4.20; 47.13)	0.090 (0.643)
		Pair 3	Snatch	521.19 ± 113.48	37.83	36.60 (13.67; 59.53)	0.006 * (1.227)
		Pair 4	Snatch Pull	518.99 ± 121.16	40.39	38.80 (19.07; 58.53	0.002 * (1.512)
		Pair 5	Back Squat	518.82 ± 128.90	42.97	38.97 (21.13; 56.81)	0.001 * (1.679)
	**Male**	SPT		Baseline			
				818,12 ± 142.13	41.03		
				Post			
		Pair 1	Muscle Snatch	822.79 ± 137.79	39.78	−4.68 (−45.08; 35.73)	0.804 (−0.074)
		Pair 2	Power Snatch	789.84 ± 137.47	39.68	28.28 (−13.89; 70.44)	0.168 (0.426)
		Pair 3	Snatch	794.05 ± 130.53	37.68	24.07 (−28.01; 76.14)	0.331 (0.294)
		Pair 4	Snatch Pull	773.47 ± 135.84	39.21	44.65 (−2.79; 92.08)	0.063 (0.598)
		Pair 5	Back Squat	785.70 ± 143.73	41.49	32.42 (−13.22; 78.05)	0.146 (0.451)
**Peak Velocity (m/s)**	**Female**	SPT		Baseline			
				1.88 ± 0.17	0.06		
				Post			
		Pair 1	Muscle Snatch	1.84 ± 0.15	0.05	0.04 (−0.01; 0.09)	0.102 (0.615)
		Pair 2	Power Snatch	1.86 ± 0.16	0.05	0.02 (−0.04; 0.08)	0.422 (0.282)
		Pair 3	Snatch	1.78 ± 0.15	0.05	0.10 (0.05; 0.16)	0.002 * (1.469)
		Pair 4	Snatch Pull	1.76 ± 0.17	0.06	0.12 (0.05; 0.20)	0.005 * (1.258)
		Pair 5	Back Squat	1.74 ± 0.21	0.07	0.14 (0.09; 0.20)	<0.001 * (2.058)
	**Male**	SPT		Baseline			
				1.76 ± 0.16	0.05		
				Post			
		Pair 1	Muscle Snatch	1.73 ± 0.19	0.05	0.04 (−0.05; 0.12)	0.378 (0.265)
		Pair 2	Power Snatch	1.68 ± 0.18	0.05	0.09 (0.03; 0.14)	0.009 * (0.910)
		Pair 3	Snatch	1.69 ± 0.17	0.05	0.07 (0.01; 0.13)	0.025 * (0.745)
		Pair 4	Snatch Pull	1.69 ± 0.14	0.04	0.07 (0.00; 0.14)	0.039 * (0.675)
		Pair 5	Back Squat	1.73 ± 0.17	0.05	0.04 (−0.01; 0.09)	0.134 (0.467)
**Mean Velocity (m/s)**	**Female**	SPT		Baseline			
				0.99 ± 0.14	0.05		
				Post			
		Pair 1	Muscle Snatch	0.97 ± 0.13	0.04	0.03 (−0.00; 0.06)	0.063 (0.719)
		Pair 2	Power Snatch	0.97 ± 0.14	0.05	0.02 (−0.01; 0.06)	0.144 (0.540)
		Pair 3	Snatch	0.93 ± 0.14	0.05	0.06 (0.02; 0.10)	0.006 * (1.228)
		Pair 4	Snatch Pull	0.92 ± 0.11	0.04	0.07 (0.03; 0.11)	0.003 * (1.372)
		Pair 5	Back Squat	0.92 ± 0.14	0.05	0.07 (0.04; 0.10)	0.001 * (1.660)
	**Male**	SPT		Baseline			
				0.89 ± 0.12	0.03		
				Post			
		Pair 1	Muscle Snatch	0.90 ± 0.10	0.03	−0.01 (−0.05; 0.04)	0.806 (−0.073)
		Pair 2	Power Snatch	0.86 ± 0.11	0.03	0.03 (−0.02; 0.08)	0.174 (0.419)
		Pair 3	Snatch	0.87 ± 0.12	0.03	0.02 (−0.04; 0.08)	0.412 (0.246)
		Pair 4	Snatch Pull	0.91 ± 0.18	0.05	−0.02 (−0.17; 0.13)	0.800 (−0.075)
		Pair 5	Back Squat	0.86 ± 0.13	0.04	0.03 (−0.02; 0.09)	0.174 (0.420)

SPT, Snatch Pull test; ROM, range of motion; *, *p* < 0.05; SD, standard deviation; SEM, standard error of the mean; MD, mean difference; CI, confidence intervals; ES, effect size.

**Table 4 healthcare-10-02499-t004:** Baseline and post-values of Snatch Pull test for the Clean and Jerk derivatives (♀♂ = 21).

Parameter	Weightlifting Derivative		Mean ± SD	SEM	MD (95% CI)	*p* (ES)
**ROM** **(cm)**	SPT		Baseline			
			106.01 ± 8.00	1.75		
			Post			
	Pair 1	Power Clean	105.77 ± 7.91	1.73	0.24 (−0.91; 1.39)	0.671 (0.094)
	Pair 2	Clean and Jerk	103.91 ± 8.88	1.94	2.10 (0.46; 3.73)	0.015 * (0.582)
	Pair 3	Clean	104.67 ± 8.77	1.91	1.34 (−0.27; 2.96)	0.098 (0.378)
	Pair 4	High Hang Clean	105.03 ± 8.98	1.96	0.98 (−0.43; 2.38)	0.164 (0.316)
	Pair 5	Hang Power Clean	104.92 ± 8.41	1.83	1.09 (−0.81; 2.99)	0.245 (0.261)
**Mean Power** **(w)**	SPT		Baseline			
			699.81 ± 176.31	38.47		
			Post			
	Pair 1	Power Clean	700.49 ± 183.15	39.97	−0.68 (−16.10; 14.74)	0.928 (−0.020)
	Pair 2	Clean and Jerk	675.26 ± 170.43	37.19	24.55 (1.65; 47.44)	0.037 * (0.488)
	Pair 3	Clean	679.59 ± 180.17	39.32	20.22 (−5.70; 46.14)	0.119 (0.355)
	Pair 4	High Hang Clean	690.40 ± 178.72	39.00	9.41 (−16.66; 35.48)	0.460 (0.164)
	Pair 5	Hang Power Clean	687.63 ± 176.81	38.58	12.18 (−13.46; 37.82)	0.334 (0.216)
**Peak Velocity (m/s)**	SPT		Baseline			
			1.75 ± 0.16	0.03		
			Post			
	Pair 1	Power Clean	1.75 ± 0.17	0.04	−0.01 (−0.05; 0.04)	0.809 (−0.054)
	Pair 2	Clean and Jerk	1.74 ± 0.18	0.04	0.01 (−0.02; 0.04)	0.456 (0.166)
	Pair 3	Clean	1.74 ± 0.18	0.04	0.01 (−0.03; 0.04)	0.712 (0.082)
	Pair 4	High Hang Clean	1.74 ± 0.16	0.03	0.01 (−0.02; 0.04)	0.511 (0.146)
	Pair 5	Hang Power Clean	1.75 ± 0.15	0.03	0.00 (−0.03; 0.04)	0.819 (0.051)
**Mean Velocity (m/s)**	SPT		Baseline			
			0.93 ± 0.11	0.02		
			Post			
	Pair 1	Power Clean	0.93 ± 0.11	0.02	0.00 (−0.02; 0.02)	0.846 (0.043)
	Pair 2	Clean and Jerk	0.90 ± 0.12	0.03	0.03 (0.00; 0.06)	0.050 (0.478)
	Pair 3	Clean	0.90 ± 0.11	0.02	0.03 (−0.00; 0.06)	0.071 (0.415)
	Pair 4	High Hang Clean	0.91 ± 0.11	0.03	0.01 (−0.02; 0.04)	0.358 (0.205)
	Pair 5	Hang Power Clean	0.91 ± 0.11	0.02	0.02 (−0.01; 0.05)	0.227 (0.272)

SPT, Snatch Pull test; ROM, range of motion; *, *p* < 0.05; SD, standard deviation; SEM, standard error of the mean; MD, mean difference; CI, confidence intervals; ES, effect size.

**Table 5 healthcare-10-02499-t005:** Baseline and post-values of Snatch Pull test for the Clean and Jerk derivatives (♀ = 9; ♂ = 12).

Parameter		Weightlifting Derivative		Mean ± SD	SEM	MD (95% CI)	*p* (ES)
**ROM (cm)**	**Female**	SPT		Baseline			
				102.14 ± 6.68	2.23		
				Post			
		Pair 1	Power Clean	102.77 ± 7.98	2.66	−0.62 (−2.78; 1.54)	0.525 (−0.222)
		Pair 2	Clean and Jerk	101.93 ± 8.16	2.72	0.21 (−2.16; 2.58)	0.843 (0.068)
		Pair 3	Clean	102.18 ± 8.70	2.90	−0.03 (−3.03; 2.96)	0.980 (−0.009)
		Pair 4	High Hang Clean	102.03 ± 8.87	2.96	0.11 (−2.42; 2.64)	0.922 (0.034)
		Pair 5	Hang Power Clean	101.47 ± 7.26	2.42	0.68 (−1.42; 2.77)	0.477 (0.248)
	**Male**	SPT		Baseline			
				108.91 ± 7.91	2.28		
				Post			
		Pair 1	Power Clean	108.03 ± 7.38	2.13	0.88 (−0.51; 2.28)	0.192 (0.401)
		Pair 2	Clean and Jerk	105.40 ± 9.44	2.73	3.51 (1.35; 5.67)	0.004 * (1.033)
		Pair 3	Clean	106.53 ± 8.72	2.52	2.38 (0.46; 4.29)	0.020 * (0.786)
		Pair 4	High Hang Clean	107.28 ± 8.74	2.52	1.63 (−0.22; 3.47)	0.079 (0.559)
		Pair 5	Hang Power Clean	107.51 ± 8.55	2.47	1.40 (−1.84; 4.64)	0.362 (0.275)
**Mean Power (w)**	**Female**	SPT		Baseline			
				536.97 ± 100.78	33.59		
				Post			
		Pair 1	Power Clean	539.52 ± 125.88	41.96	−2.56 (−27.66; 22.55)	0.820 (−0.078)
		Pair 2	Clean and Jerk	539.18 ± 132.05	44.02	−2.21 (−33.10; 28.63)	0.873 (−0.055)
		Pair 3	Clean	533.22 ± 130.91	43.64	3.74 (−26.22; 33.70)	0.781 (0.096)
		Pair 4	High Hang Clean	542.37 ± 125.61	41.87	−5.40 (−27.63; 16.83)	0.591 (−0.187)
		Pair 5	Hang Power Clean	528.48 ± 120.75	40.30	8.49 (−15.44; 32.41)	0.437 (0.273)
	**Male**	SPT		Baseline			
				821.94 ± 105.66	30.50		
				Post			
		Pair 1	Power Clean	821.21 ± 111.21	32.10	0.73 (−22.23; 23.69)	0.945 (0.020)
		Pair 2	Clean and Jerk	777.33 ± 116.69	33.68	44.62 (13.47; 75.76)	0.009 * (0.910)
		Pair 3	Clean	789.37 ± 126.03	36.38	32.58 (−9.58; 74.73)	0.117 (0.491)
		Pair 4	High Hang Clean	801.42 ± 123.38	35.62	20.53(−24.59; 65.64)	0.338 (0.289)
		Pair 5	Hang Power Clean	806.99 ± 99.85	28.82	14.95 (−30.17; 60.07)	0.481 (0.211)
**Peak Velocity (m/s)**	**Female**	SPT		Baseline			
				1.80 ± 0.13	0.04		
				Post			
		Pair 1	Power Clean	1.80 ± 0.15	0.05	−0.01 (−0.10; 0.05)	0.795 (−0.090)
		Pair 2	Clean and Jerk	1.82 ± 0.12	0.04	−0.02 (−0.06; 0.02)	0.231 (−0.081)
		Pair 3	Clean	1.81 ± 0.15	0.05	−0.01 (−0.07; 0.05)	0.725 (−0.121)
		Pair 4	High Hang Clean	1.79 ± 0.15	0.05	0.01 (−0.06; 0.07)	0.849 (0.066)
		Pair 5	Hang Power Clean	1.77 ± 0.16	0.05	0.03 (−0.04; 0.10)	0.377 (0.312)
	**Male**	SPT		Baseline			
				1.72 ± 0.17	0.05		
				Post			
		Pair 1	Power Clean	1.72 ± 0.18	0.05	−0.00 (−0.08; 0.07)	0.903 (−0.036)
		Pair 2	Clean and Jerk	1.68 ± 0.20	0.06	0.04 (−0.01; 0.08)	0.089 (0.539)
		Pair 3	Clean	1.70 ± 0.19	0.05	0.02 (−0.03; 0.07)	0.437 (0.233)
		Pair 4	High Hang Clean	1.70 ± 0.16	0.05	0.01 (−0.03; 0.05)	0.459 (0.222)
		Pair 5	Hang Power Clean	1.73 ± 0.15	0.04	−0.02 (−0.06; 0.03)	0.492 (−0.205)
**Mean Velocity (m/s)**	**Female**	SPT		Baseline			
				0.96 ± 0.12	0.04		
				Post			
		Pair 1	Power Clean	0.96 ± 0.12	0.04	0.00 (−0.04; 0.04)	0.901 (0.043)
		Pair 2	Clean and Jerk	0.96 ± 0.12	0.04	0.01 (−0.04; 0.06)	0.773 (0.099)
		Pair 3	Clean	0.95 ± 0.11	0.04	0.02 (−0.04; 0.07)	0.493 (0.239)
		Pair 4	High Hang Clean	0.96 ± 0.11	0.04	−0.00 (−0.04; 0.04)	0.947 (−0.023)
		Pair 5	Hang Power Clean	0.94 ± 0.11	0.04	0.02 (−0.03; 0.07)	0.322 (0.352)
	**Male**	SPT		Baseline			
				0.90 ± 0.10	0.03		
				Post			
		Pair 1	Power Clean	0.90 ± 0.10	0.03	0.00 (−0.02; 0.03)	0.890 (0.041)
		Pair 2	Clean and Jerk	0.86 ± 0.10	0.03	0.05 (0.01; 0.08)	0.011 * (0.876)
		Pair 3	Clean	0.86 ± 0.10	0.03	0.04 (−0.01; 0.08)	0.091 (0.535)
		Pair 4	High Hang Clean	0.88 ± 0.11	0.03	0.02 (−0.02; 0.07)	0.282 (0.326)
		Pair 5	Hang Power Clean	0.89 ± 0.11	0.03	0.01 (−0.03; 0.06)	0.489 (0.207)

SPT, Snatch Pull test; ROM, range of motion; *, *p* < 0.05; SD, standard deviation; SEM, standard error of the mean; MD, mean difference; CI, confidence intervals; ES, effect size.

## Data Availability

The data presented in this study are available upon request from the corresponding author.

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
