# Peer review of "Comparison between Olympic Weightlifting Lifts and Derivatives for External Load and Fatigue Monitoring"

_healthcare, 2022, doi:10.3390/healthcare10122499_

Round 1

Reviewer 1 Report

I think that some references too old

Reviewer 2 Report

See comments attached!

Reviewer 3 Report

The aim of the present study was to compare the fatigue prompted by the Clean & Jerk, the Snatch, and their derivative exercises (Snatch, Snatch Pull, Muscle Snatch, Power Snatch, and Back Squat; Clean & Jerk, Power Clean, Clean, High Hang Clean, Hang Power Clean) for the total, male and female participants, respectively. The manuscript covers the important problem of optimization the training in the Olympic Weightlifting. However, it is unclear if the problem is still relevant. The authors refer to many papers published more than 10 years ago. It is possible that the approach to studying this problem is outdated. The manuscript has flaws in the methodology, due to which the result of the study could not be properly assessed. First, the purpose of the study should be clarified. Then a proper study design must be organized.

The Materials and Methods section should be modified. It is necessary clearly present the methodology and repetition should be avoided. The study design is unclear. DOI address: dx.doi.org/10.17504/protocols.io.n92ldzxq8v5b/v1 failed to visit.

What were the reasons of exercise selections (p.2.3)? The exercises were only listed without explanation. Possible randomization of exercises was not noted. The standardization of the load in different subjects is also not noted.

The paragraph 2.4 should clearly present the methods of performance assessment; all discussion is better presented in the introduction. What parameters were measured? How was fatigue assessed? It seems that in the male group, the exercise did not cause sufficient fatigue.

The abstract should more clearly state the results obtained during the study. The aim of the study in the abstract is different from those in the maintext.

The discussion is weak. It does not present the results with similar exercise protocols.

The conclusion that “this intervention confirmed the hypothesis that when volume and intensity are equated, there are differences between fatigue induced by various OW exercises” is irrelevant because the load was not standardized. The conclusion that “females seem more sensible to fatigue in Snatch derivatives” (as well in C&J derivatives) is not supported with the results as results were not compared across gender groups.

It is unclear the practical outcome of the study, as the individual effects were not assessed.

My overall comment: The manuscript in its present form is not ready for publishing. Article has serious flaws, research not conducted correctly.

Round 2

Reviewer 3 Report

Dear Authors,

The manuscript has been slightly modified. However it still has flaws in the methodology, due to which the result of the study could not be properly assessed. Correct terminology should be used to explain muscle function during exercise. The authors mainly performed external load monitoring rather than fatigue assessment.

I would recommend Halson SL (2014) for a better understanding of muscle function assessment.

Halson SL. Monitoring training load to understand fatigue in athletes. Sports Med. 2014 Nov;44 Suppl 2(Suppl 2):S139-47. doi: 10.1007/s40279-014-0253-z. PMID: 25200666; PMCID: PMC4213373.

Since no fatigue assessment was provided, the title of the manuscript should be modified.

The relevance of the study is also questionable. No changes in the list of references have been made.

It is not clear why the purpose of the study is presented differently in parts of the manuscript.

The Authors presented the main outcome in a form of equations in the Discussion section. Given the flaws in the terminology, it cannot be considered a “Fatigue Impact Quantification”, as quantification has not been proven. The Discussion section has not been modified enough.

The Conclusion section has not changed. The conclusions are not supported by the results. The previous comment (The conclusion that “females seem more sensible to fatigue in Snatch derivatives” (as well in C&J derivatives) is not supported with the results as results were not compared across gender groups) was not considered.

My overall comment: The manuscript in its present form is not ready for publishing. Article has serious flaws, research not conducted correctly.
